# Optimal Treatment Strategy for Oligo-Recurrence Lung Cancer Patients with Driver Mutations

**DOI:** 10.3390/cancers16020464

**Published:** 2024-01-22

**Authors:** Taimei Tachibana, Yosuke Matsuura, Hironori Ninomiya, Junji Ichinose, Masayuki Nakao, Sakae Okumura, Makoto Nishio, Norihiko Ikeda, Mingyon Mun

**Affiliations:** 1Department of Thoracic Surgical Oncology, Cancer Institute Hospital, Japanese Foundation for Cancer Research, Tokyo 135-8550, Japan; taimei.tachibana@jfcr.or.jp (T.T.); junji.ichinose@jfcr.or.jp (J.I.); masayuki.nakao@jfcr.or.jp (M.N.); sokumura@jfcr.or.jp (S.O.); mingyon.mun@jfcr.or.jp (M.M.); 2Department of Surgery, Tokyo Medical University, Tokyo 160-8402, Japan; ikeda-n@tokyo-med.ac.jp; 3Division of Pathology, Cancer Institute Hospital, Japanese Foundation for Cancer Research, Tokyo 135-8550, Japan; hironori.ninomiya@jfcr.or.jp; 4Department of Pathology, Cancer Institute Hospital, Japanese Foundation for Cancer Research, Tokyo 135-8550, Japan; 5Department of Thoracic Medical Oncology, Cancer Institute Hospital, Japanese Foundation for Cancer Research, Tokyo 135-8550, Japan; mnishio@jfcr.or.jp

**Keywords:** lung cancer, oligo-recurrence, driver mutation, local therapy, molecular targeted therapy

## Abstract

**Simple Summary:**

Molecular targeted therapies are very effective as a treatment for postoperative recurrence of non-small cell lung cancer (NCSLC) with driver mutations. On the other hand, the efficacy of local therapies for oligo-recurrence has also been reported. To investigate the optimal treatment strategy for oligo-recurrence in NSCLC patients with driver mutations, we retrospectively evaluated 66 NSCLC patients with driver mutations who were treated with local or molecular targeted therapies as the initial treatment after recurrence. Patients treated with local therapies as a first-line treatment did not show statistically significant differences in post-recurrence survival and progression-free survival compared with those treated with molecular targeted therapies. However, local therapies as an initial treatment should be considered preferably, as they can be curative after recurrence and can delay the start of the administration of molecular targeted therapies.

**Abstract:**

Background: The efficacy of local therapies for lung cancer patients with postoperative oligo-recurrence has been reported. However, whether local therapies should be chosen over molecular targeted therapies for oligo-recurrence patients with driver mutations remains controversial. Therefore, we aimed to investigate the optimal initial treatment strategy for oligo-recurrence in lung cancer patients with driver mutations. Methods: Among 2152 patients with lung adenocarcinoma who underwent surgical resection at our institute between 2008 and 2020, 66 patients with driver mutations who experienced cancer oligo-recurrence after surgery and were treated with local or molecularly targeted therapy as an initial therapy after recurrence were evaluated. Oligo-recurrence was characterized by the presence of 1 to 3 recurrent lesions. These patients were investigated, focusing on their post-recurrence therapies and prognoses. Results: The median follow-up period was 71 months. Local and molecular targeted therapies were administered to 41 and 25 patients, respectively. The number of recurrence lesions tended to be lower in the initial local therapy group than in the molecular targeted therapy group. In the initial local therapy group, 23 patients (56%) subsequently received molecular targeted therapies. The time from recurrence to the initiation of molecular targeted therapy was significantly longer in the local therapy group than in the molecular targeted therapy group (*p* < 0.001). There was no significant difference in post-recurrence overall survival (hazard ratio, 1.429; 95% confidence interval, 0.701–2.912; log-rank, *p* = 0.324) and post-recurrence progression-free survival (hazard ratio, 0.799; 95% confidence interval, 0.459–1.390; log-rank, *p* = 0.426) in the initial local ablative therapy group compared with the initial molecular targeted therapy group. Conclusions: Local therapies as a first-line treatment did not show statistically significant differences in post-recurrence survival or progression-free survival compared with molecular targeted therapies. However, local therapies as an initial treatment should be considered preferably, as they can cure the recurrence and can delay the start of administration of molecular targeted therapies.

## 1. Introduction

The discovery of driver mutations has markedly changed the treatment for non-small cell lung cancer (NSCLC) [1,2]. Specific molecular targeted therapies (MTTs) have achieved substantial therapeutic effects in patients with driver mutations. In particular, advanced NSCLC patients with driver mutations, including epidermal growth factor receptor (EGFR) mutations and anaplastic lymphoma kinase (ALK) rearrangement, can benefit from treatment with MTTs [1,2,3]. Furthermore, MTTs are preferentially used for NSCLC patients with driver mutations who develop postoperative recurrence because postoperative recurrence is considered to be a systemic disease [4].

However, some NSCLC patients have oligo-recurrence [5]. The concept of oligo-recurrence was originally proposed in 1995 as an intermediate state of cancer spread between localized disease and widespread disease and was defined as (i) one to several distant metastases or recurrences in one to several organs (usually one), (ii) the primary site of cancer is controlled, (iii) one to several distant metastases or recurrences can be treated with local ablative therapies (LATs), and (iv) no distant metastases or recurrences other than (iii). Oligo-recurrence differs from oligo-metastases in that the primary tumor is under control, and all grossly recurrent tumors can be treated with LATS [5,6]. Oligo-recurrence of NSCLC is also often treatable by LATs, including surgical resection and definitive radiotherapy [7,8].

In 2019, a consensus report was published from the United States and Europe on the definition and classification of oligo-metastases, including oligo-recurrence in NSCLC. Since then, the treatment strategies for oligo-recurrence have attracted worldwide attention, including sessions at many conferences and clinical trials. However, they exclude patients with driver mutations, as reported by Iyengar et al. [9]. Moreover, a previous study suggested that the use of MTTs for oligo-recurrence in NSCLC patients with EGFR mutations did not improve prognosis after recurrence [10]. However, very few studies to date have directly compared the efficacy of LAT and MTT in NSCLC patients with driver mutations who have oligo-recurrence, and whether LAT should be chosen over MTT in this population remains controversial. Therefore, in the present study we aim to investigate the optimal initial treatment strategy for oligo-recurrence in NSCLC patients with driver mutations.

## 2. Materials and Methods

### 2.1. Study Design and Patient Selection

This retrospective study was approved by the Institutional Review Board of Cancer Institute Hospital, Japanese Foundation for Cancer Research in Tokyo, Japan (study approval no.: 2021-GA-1304) and was conducted in accordance with the amended Declaration of Helsinki. The need for written informed consent was waived because of the retrospective nature of the study and the anonymity of the subjects.

Of the 2152 consecutive lung adenocarcinoma patients who underwent surgical resection at Cancer Institute Hospital, Japanese Foundation for Cancer Research, Tokyo, Japan, between 2008 and 2020, patients with EGFR mutations or ALK rearrangement who underwent lobectomy or more extensive pulmonary resection with mediastinal lymphadenectomy, and who developed postoperative recurrence of any type were analyzed. Patients who did not have recurrence (*n* = 1767), did not have EGFR and ALK mutations (*n* = 1130), were treated with preoperative induction therapies (*n* = 13), underwent R1 or R2 resection (*n* = 71), had recurrence at the resected stump (*n* = 3), did not undergo treatment after recurrence (*n* = 16), or were lost to follow-up or whose data were unavailable (*n* = 32) were excluded from the study. Initial recurrence was classified into two categories based on the recurrence site, i.e., locoregional recurrence and distant recurrence. Locoregional recurrence was defined as the presence of a tumor within the ipsilateral hemithorax and regional lymph nodes, such as the hilar, mediastinal, and cervical lymph nodes. Distant recurrence was defined as the presence of a tumor in the contralateral lung or outside the hemithorax. Furthermore, recurrence types were classified into two categories, i.e., oligo-recurrence and poly-recurrence. Oligo-recurrence was defined as one to three locoregional and/or distant recurrences as the first recurrence, with a regional lymph node station classified as a single organ. Poly-recurrence was defined as other types of recurrence. Patients with poly-recurrence were excluded from this study. A total of 66 patients with available data were finally evaluated. A flowchart showing the patient selection protocol is shown in Figure 1.

Using medical records, data on the following clinicopathological characteristics were collected: age, sex, smoking status, pathological stage, tumor size, types of driver mutations, degree of histological differentiation, lymph-vascular invasion, visceral pleural invasion, intrapulmonary metastasis, time to recurrence (TTR) after surgery, time from recurrence to treatment, time from recurrence to initiation of tyrosine kinase inhibitor (TKI) treatment, recurrence pattern, number of recurrence lesions, and initial treatment post-recurrence. Pathological staging was performed according to the 8th edition of the Union for International Cancer Control guidelines [11]. The degree of histological differentiation of the resected specimens was determined according to the 4th edition of the World Health Organization classification system [12].

### 2.2. Postoperative Surveillance and Definition of Recurrence Patterns and Initial Therapies Post-Recurrence

All patients underwent chest and abdominal computed tomography (CT) and blood examinations twice a year for two years after the initial surgery and once a year thereafter [13]. Patients who presented with any symptom or sign of recurrence in these examinations underwent further evaluation, including brain magnetic resonance imaging and positron emission tomography. Recurrence was identified on the basis of the imaging findings and was histologically and/or cytologically confirmed when possible. Recurrence sites and the number of recurrence lesions were reviewed. The initial treatment post-recurrence was classified into two categories, i.e., LAT and MTT. LAT was defined as treatments performed with curative intent, including complete surgical resection, stereotactic ablation radiotherapy, cerebral stereotactic radiosurgery, other radical radiation therapies of at least 45 Gy, and concurrent chemoradiotherapy. MTT included any kind of EGFR-TKI or ALK-TKI treatment that could be used in clinical practice. In all cases, the course of treatment for each patient with recurrence was decided after a comprehensive evaluation of recurrence patterns, performance status, and patients’ social backgrounds and preferences at a conference held by the multidisciplinary thoracic oncology team at our institution, including respiratory medicine, respiratory surgery, and radiology. Stereotactic radiosurgery is the preferred option for cerebral oligo-recurrence [14]. All patients with recurrence underwent head, chest, and abdominal CT and blood examinations every three months after the diagnosis of recurrence or the start of post-recurrence therapy. Disease progression was determined in accordance with the Response Evaluation Criteria in Solid Tumor, ver. 1.1 (RECIST 1.1). The prognoses of the two groups of patients who received LATs or MTTs as the post-recurrence initial therapy were compared. Moreover, recurrence patterns and prognoses, including the course of treatment after the initial therapy, were investigated. Prognoses after recurrence were assessed using post-recurrence overall survival (PR-OS) and post-recurrence progression-free survival (PR-PFS) rates.

### 2.3. Detection of Driver Mutations

EGFR mutational status was determined using Cobas^®^ EGFR Mutation Test v2 (Roche Molecular Systems, Inc., Pleasanton, CA, USA) [15]. ALK rearrangement was evaluated with immunohistochemical (IHC) analysis using ALK Detection Kits^®^ (Nichirei Bioscience, Tokyo, Japan) [15]. ALK positivity on IHC analysis was defined as tumor cell staining of more than 80%. Positive cases were further examined for confirmation using Vysis^®^ ALK Break-Apart fluorescence in situ hybridization (FISH) Probe Kits (Abbott Molecular, Chicago, IL, USA). ALK rearrangement was considered to be positive when more than 15% of the tumor cells demonstrated split signals or single red signals [16]. All FISH and IHC analyses were performed using 4-µm-thick formalin-fixed paraffin-embedded tissue samples prepared from the surgically resected specimens. The results were evaluated by two experienced pulmonary pathologists (H.N. and Y.I.).

### 2.4. Statistical Analysis

Between-group comparisons of patient characteristics were performed using the Mann–Whitney U test for continuous variables and Fisher’s exact test for categorical variables. Survival curves were plotted using the Kaplan–Meier method, and compared using the log-rank test. Data of patients who were alive on 30 November 2023 were censored for survival analysis. TTR was calculated as the time from the date of surgery for resection of the primary lesion to the date of diagnosis of recurrence. Time from recurrence to treatment was calculated as the time from the date of diagnosis of recurrence to the date of initial treatment post-recurrence. Time from recurrence to initiation of TKI treatment was calculated as the time from the date of diagnosis of recurrence to the date of initiation of TKI treatment. PR-OS was calculated as the time from the date of disease recurrence to the date of death from any cause. PR-PFS was calculated as the time from the date of disease recurrence to the date of disease progression, death, or the last contact/follow-up if patients remained disease-free. Statistical significance was set at a *p*-value of less than 0.05. All statistical analyses were performed using the SPSS statistical software program (version 27.0; DDR3 RDIMM, SPSS Inc., Chicago, IL, USA).

## 3. Results

### 3.1. Patient Characteristics

The clinicopathological characteristics of the enrolled patients are summarized in Table 1. LATs or MTTs were administered as initial post-recurrence therapies to 41 (initial LAT group) and 25 (initial MTT group) patients. There was a significant difference in pathological stage between the two groups of patients, with the initial MTT group having more advanced disease. The number of recurrence lesions tended to be lower in the initial LAT group than in the initial MTT group.

Comparison of the sites of initial recurrence after the initial treatment in patients with oligo-recurrence with driver mutations are summarized in Table 2. In the initial LAT group, 23 (56%) patients were also treated with MTTs, and in the initial MTT group, 6 (24%) patients received LATs as an additional therapy. The median time from recurrence to initiation of TKI treatment was longer in the initial LATs group than in the initial MMTs group (37 months vs. 1 month, *p* < 0.001). However, the final number of MTTs used was not significantly different between the two groups. In the initial MTT group, 21 patients were EGFR-positive, and 4 were ALK-positive; the TKIs used in the EGFR-positive patients were first-generation in 16 patients, second-generation in 1 patient, and third-generation in 4 patients. In addition, four ALK-positive patients were treated with alectinib and one with crizotinib. Gefitinib was usually administered at a dose of 250 mg/day daily, erlotinib at a dose of 150 mg/day daily, afatinib at a dose of 40 mg/day daily, and osimertinib at a dose of 80 mg/day daily. Alectinib was usually administered at a dose of 600 mg/day daily and crizotinib at a dose of 500 mg/day daily. The dose was reduced or discontinued at the attending physician’s discretion regarding toxicity.

### 3.2. Prognostic Analyses

The median follow-up period after surgery was 71 months (range, 20–172 months). Kaplan–Meier curves comparing the two groups for PR-OS and PR-PFS are shown in Figure 2 and Figure 3, respectively. There was no significant difference in PR-OS (HR, 1.429; 95% CI, 0.701–2.912; log-rank, *p* = 0.324) and PR-PFS (HR, 0.799; 95% CI, 0.459–1.390; log-rank, *p* = 0.426) between the initial LAT group and the initial MTT group.

Similar analyses were conducted by initial recurrence site (locoregional recurrence vs. distant recurrence), no significant difference was found in either PR-OS (Figure 4, HR, 1.333; 95% CI, 0.636–2.795; log-rank, *p* = 0.445)) or PR-PFS (Figure 5, HR, 0.908; 95% CI, 0.523–1.575; log-rank, *p* = 0.730).

## 4. Discussion

### 4.1. Primary Findings

The choice of LAT over MTT for NSCLC patients with driver mutations who have oligo-recurrence remains controversial. Our results demonstrated that LAT does not prolong either PR-OS or PFS of patients compared with MTT, and it was not possible to show which was more effective compared with TKI. However, to our knowledge, the present study is one of the few studies performed to date investigating treatment strategies for lung cancer patients with driver mutations who have developed oligo-recurrence.

### 4.2. Post-Recurrence Therapies for Oligo-Recurrence Patients with Driver Mutations

LATs have been shown to improve the prognosis of both advanced NSCLC patients with oligo-metastasis who do not show progression after the first-line systemic therapy and those who develop postoperative oligo-recurrence [17,18,19]. It is well known that stereotactic body radiation therapy (SBRT) has shown good overall survival, and stereotactic radiosurgery (SRS) or stereotactic radiotherapy (SRT) has shown good prognosis for oligo-metastases in the brain only [20,21]. On the other hand, the efficacy of MTTs for postoperative recurrence of NSCLC has been reported [22,23]. The 5-year PFS and OS rates of patients with postoperative recurrence after the initiation of EGFR-TKIs were reported to be 12.9% and 51.5%, respectively, which is the standard of care for the initial MTT group in this study [24]. However, the usefulness of LATs for NSCLC patients with driver mutations has not been adequately investigated. The Randomized Phase III Trial of First-Line Tyrosine Kinase Inhibitor With or Without Radiotherapy for Synchronous Oligometastatic EGFR-Mutated NSCLC demonstrated the effectiveness of radiation therapy (RT) [25]. A randomized phase II trial of osimertinib, a representative EGFR-tyrosine kinase inhibitor, with or without LAT for patients harboring EGFR mutations with oligo-metastatic NSCLC is currently ongoing [26]. However, NSCLC patients with driver mutations are often excluded from participating in clinical trials on oligo-metastasis [27]. This is because most of the driver mutations are adenocarcinomas, and furthermore, MTTs can have substantial therapeutic effects on patients with driver mutations, which are different from the effects on NSCLC patients without driver mutations [1,2,3]. Therefore, it may be difficult to conduct prospective clinical trials regarding an optimal treatment strategy for NSCLC patients with driver mutations who have postoperative oligo-recurrence. We found only 1 study that directly compared the efficacy of LATs and MTTs on oligo-recurrence in patients with EGFR-mutated NSCLC, but this previous study also found no significant difference in PR-OS between LAT and MTT [28].

### 4.3. Interpretation of the Study Findings

The main findings of this study are as follows. First, there was no significant difference between the initial LAT group and the initial MTT group in both PR-OS and PR-PFS, and it is, hence, not possible to conclude which treatment will result in a more favorable prognosis. Second, in this study, 29 patients (71%) in the initial LAT group were eventually treated with MTTs. However, nine patients (22%) in the initial LAT group had no recurrence after treatment and were cured. Finally, LAT as initial treatment delayed the start of MTT. Although this would delay the development of resistance to MTT, there was no significant difference in the number of MTTs ultimately used. Based on these results, reading the PR-PFS survival curve, it appears that metachronous recurrence occurred in the initial LAT group for about two years after the start of treatment, but the combination of MTT suppressed these foci. As a result, during the PR-OS observation period, the survival curve of the initial LAT group was upward compared to that of the initial MTT group.

LAT has the advantage of reducing the high medical costs of TKI treatment, which is favorable from the point of view of the patients and health care economics. LAT also has the advantage of avoiding side effects by avoiding MTT and maintaining quality of life in the meantime. Therefore, our results imply that a combination of LATs and MTTs may be a promising treatment strategy for NSCLC patients with driver mutations who develop oligo-recurrence, but that LATs should be considered first.

### 4.4. Study Limitations

This study has some limitations. First, this was a retrospective observational study at a single institute. The number of cases enrolled was relatively small, which in turn affected the statistical robustness. Second, the criterion of one to three lesions for oligo-recurrence was unique to this study. Regarding oligo-metastasis, the definition has been formulated by a multidisciplinary consensus statement, which is a maximum of five metastases and three organs that can be radically treated for all tumor sites [29]. No similar consensus has been reached on oligo-recurrence, and the definition of oligo-recurrence is not uniform [5,6,10,17]. However, the European Society for Therapeutic Radiology and Oncology, the European Organization for Research and Treatment of Cancer, and the American Society for Radiation Oncology have jointly published a consensus report on the definition and classification of oligo-metastases and therapeutic strategies for oligo-metastases based on this consensus definition and classification are being developed [30]. At the consensus meeting, most researchers agreed that the criterion for oligo-metastases should be a maximum of two metastatic organs and three metastases [31]. Third, the post-recurrence treatment strategy was not standardized among patients because of the retrospective study design. In fact, some patients with driver mutations were treated without LAT despite oligo-recurrence. Finally, we did not investigate other gene alterations (e.g., ROS1, RET, BRAF, and MET), which could have also affected the study results. However, the use of broad genomic sequencing may not result in a more favorable prognosis because no specific inhibitors for these rare driver mutations are currently available [32]. Further large clinical studies are needed to overcome these limitations. In addition, our findings may be modified with the development of new agents.

## 5. Conclusions

LAT as a first-line treatment did not show statistically significant differences in post-recurrence survival or progression-free survival compared with molecular targeted therapies. However, local therapies as an initial treatment may be preferable, as they may be effective in curing recurrence and delay the start of administration of molecular targeted therapies.

## Figures and Tables

**Figure 1 cancers-16-00464-f001:**
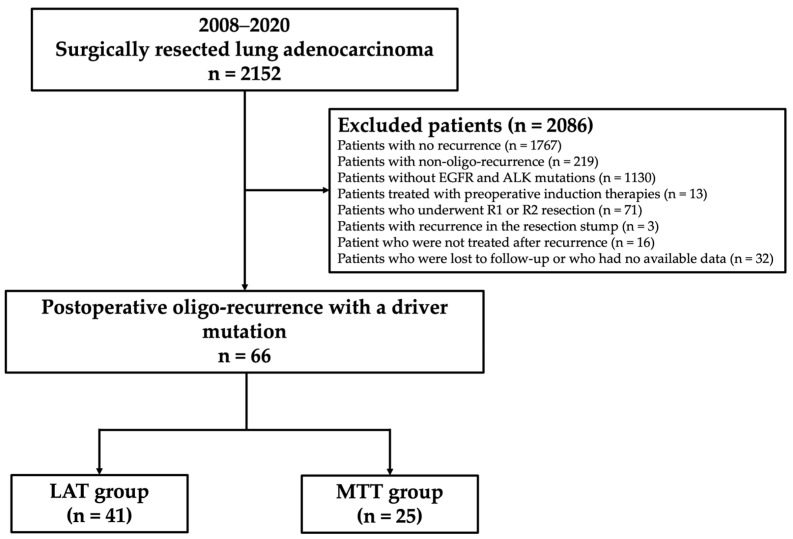
Study design and flow chart of this study. LAT, local ablative therapy; MTT, molecular targeted therapy; IQR, interquartile range; EGFR, epidermal growth factor receptor; ALK, anaplastic lymphoma kinase.

**Figure 2 cancers-16-00464-f002:**
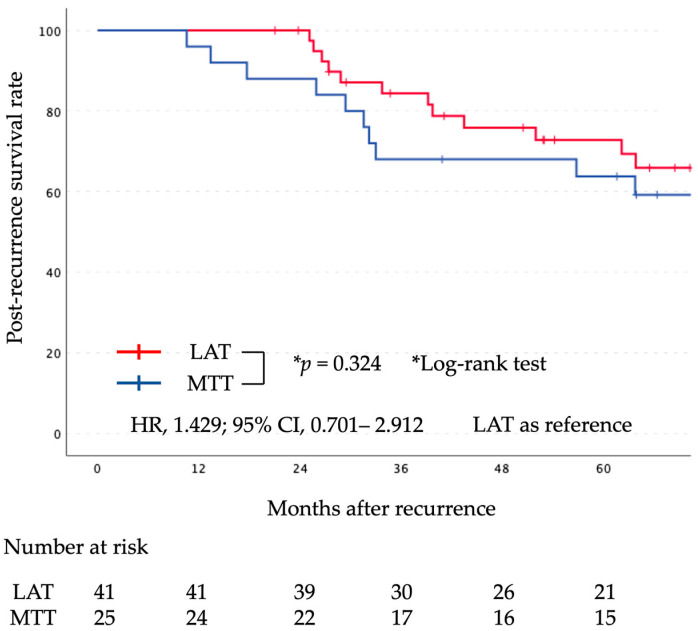
Graph indicating the PR-OS for the LAT and MTT groups. There was no statistically significant difference between the two groups. LAT, local ablative therapy; MTT, molecular targeted therapy; PR-OS, post-recurrence overall survival.

**Figure 3 cancers-16-00464-f003:**
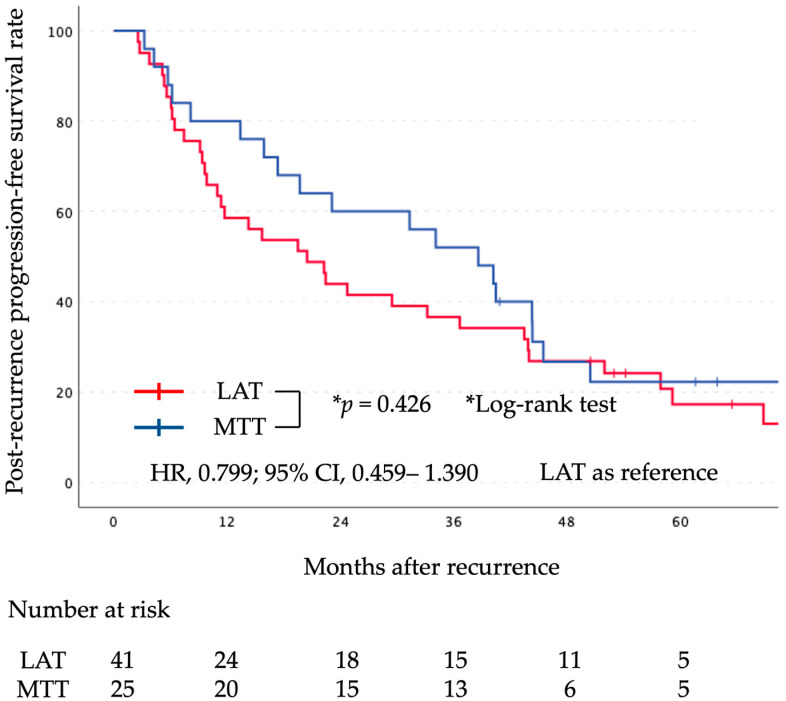
Graph indicating PR-PFS for the LAT and MTT groups. There was no statistically significant difference between the two groups. LAT, local ablative therapy; MTT, molecular targeted therapy; PR-PFS, post-recurrence progression-free survival.

**Figure 4 cancers-16-00464-f004:**
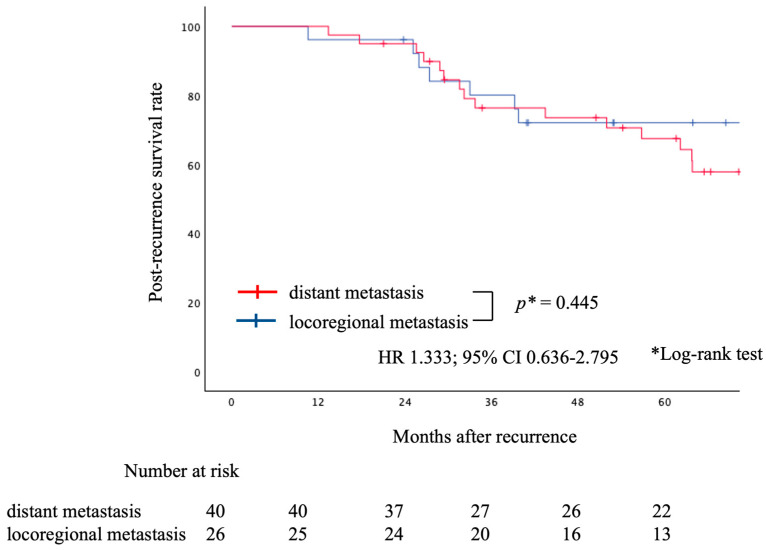
Graph indicating PR-OS for the distant and locoregional metastasis groups. There was no statistically significant difference between the two groups. PR-OS, post-recurrence overall survival.

**Figure 5 cancers-16-00464-f005:**
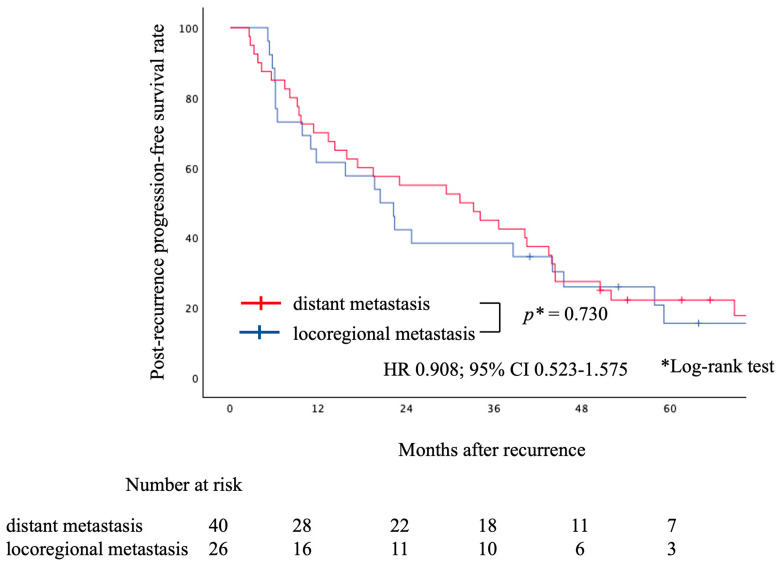
Graph indicating PR-PFS for the distant and locoregional metastasis groups. There was no statistically significant difference between the two groups. PR-PFS, post-recurrence progression-free survival.

**Table 1 cancers-16-00464-t001:** Characteristics of patients with driver mutations who developed postoperative recurrence, grouped by their initial treatment method.

Variable	Initial LAT (*n* = 41)	Initial MTT (*n* = 25)	*p*-Value *
Age, years			
median (IQR)	66 (61–73)	65 (56–75)	0.431
Sex,			
male, *n* (%)	16 (39)	11 (44)	0.690
Smoking status,			
pack-years, median (IQR)	2 (0–550)	5 (0–197)	0.951
Pathological stage **,			
I/II/III, *n* (%)	18 (44)/8 (20)/15 (37)	3 (12)/8 (32)/14 (56)	0.026
Tumor size, mm,			
median (IQR)	28 (23–36)	28 (20–43)	0.776
Driver mutation,			
EGFR/ALK, *n* (%)	58 (88)/8 (12)	21 (84)/4 (16)	0.451
Histological differentiation, well/moderate/poor, *n* (%)	2 (5)/32 (78)/7 (17)	4 (16)/18 (72)/3 (12)	0.293
Lymphatic invasion, *n* (%)	24 (59)	16 (64)	0.659
Vascular invasion, *n* (%)	30 (73)	20 (80)	0.530
Visceral pleural invasion, *n* (%)	23 (56)	14 (56)	0.994
Intrapulmonary metastasis, *n* (%)	7 (17)	7 (28)	0.292
Initial recurrence site,			
locoregional/distant/both, *n* (%)	18 (43)/21 (51)/2 (5)	8 (32)/15 (60)/2 (8)	0.598
Number of recurrence lesions, *n* (%)			
1	32 (78)	5 (20)	
2	7 (17)	14 (56)	
3	2 (5)	6 (24)	<0.001
Time from recurrence to treatment, months, median (IQR)	1 (0–2)	1 (0–6)	0.658
Time from recurrence to initiation of MTT, months, median (IQR)	33 (12–53)	1 (1–7)	<0.001

* *p*-values were calculated using Fisher’s exact test or the Mann–Whitney U test. ** The 8th edition of the Union for International Cancer Control guidelines. LAT, local ablative therapy; MTT, molecular targeted therapy; IQR, interquartile range; EGFR, epidermal growth factor receptor; ALK, anaplastic lymphoma kinase.

**Table 2 cancers-16-00464-t002:** Comparison of initial recurrence sites in patients with driver mutations who developed oligo-recurrence, grouped by their initial treatment method.

Initial Recurrence Site **	Initial LAT (*n* = 41)	Initial MTT (*n* = 25)	*p*-Value *
Locoregional recurrence, *n*	20	10	0.487
Regional lymph node stations	12	6	
Ipsilateral lung	8	4	
Distant recurrence, *n*	23	17	0.337
Brain	15	2	
Contralateral lung	3	7	
Liver	1	4	
Bone	7	3	
Adrenal gland	0	2	
Initial therapy post-recurrence, *n* (%)			
Surgical resection	15 (37)		
Cerebral stereotactic radiosurgery	14 (34)		
Other radiotherapies	14 (34)		
EGFR-MTTs		21 (84)	
ALK-MTTs		4 (16)	
Subsequent administration of LATs, *n* (%)		6 (24)	
Subsequent administration of MTTs, *n* (%)	23 (56)		
Number of MTTs used, median (range)	1 (0–4)	1 (1–3)	0.249

* *p*-values were calculated using Fisher’s exact test or the Mann–Whitney U test. ** Includes overlapping cases. LAT, local ablative therapy; MTT, molecular targeted therapy; EGFR, epidermal growth factor receptor; ALK, anaplastic lymphoma kinase.

## Data Availability

The data of this study are available from the authors upon reasonable request.

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
