# Peer review of "Optimal Treatment Strategy for Oligo-Recurrence Lung Cancer Patients with Driver Mutations"

_cancers, 2024, doi:10.3390/cancers16020464_

Round 1
Reviewer 1 Report
Comments and Suggestions for Authors
Thank you for submitting this interesting and informative manuscript to Cancers. I was pleased to receive it as a reviewer.
While your manuscript provides valuable insights into an important clinical topic, there are certain areas that could be refined to further enhance the quality and impact of the work. Here are some respectful suggestions that could potentially improve the paper if you choose to implement them:
Introduction
- Consider expanding the background on the biological rationale behind the oligo-recurrence phenomenon (e.g., tumour biology, metastatic inefficiency, mutations enabling oligo-metastatic growth, etc). This could better frame why an oligo-recurrent state may exist separately from widespread disease.
- Consider expanding on prior studies of localized therapies for oligo-recurrence in NSCLC without driver mutations to better contextualize this work's focus on the mutant population.
Methods
- Consider providing more specifics on patient selection criteria. For instance, were baseline characteristics balanced between groups? Were any statistical methods used to control confounding factors? Describing these details would strengthen confidence in the comparison.
- Consider elaborating on how post-recurrence treatments were selected: was it standardized or physician-dependent? Were the cases discussed at a multidisciplinary tumour board? A few comments on factors influencing choices would be informative.
- Consider providing more specifics on the systemic therapy regimens administered (e.g., tyrosine kinase inhibitor generation, doses, and duration) so readers can understand the real-world treatment landscape.
Results
- Consider adding subgroup outcome analyses by oligo-recurrence site (locoregional vs distant) or mutation type (EGFR vs ALK) to uncover informative trends within this heterogeneous population.
- Consider reporting on sites of eventual progression when described to unveil differences between upfront systemic versus ablative therapy.
Discussion
- Consider further comparing survival curves and discussing similarities/differences in more depth. Speculate on factors potentially influencing the patterns observed.
- Consider commenting on whether delayed targeted therapy initiation may improve quality of life in the interim by avoiding side effects. This could be another benefit to the sequencing approach proposed.
- Consider comparing your survival outcomes to existing benchmarks for NSCLC with driver mutations (historical controls). This could better contextualise your study results.
- Consider discussing whether any prognostic clinical or molecular factors may explain response to ablation versus TKIs. This may reveal predictors to personalize treatment.
Overall, these suggestions aim to enhance the manuscript's quality and impact for clinicians and researchers considering adoption of ablative and systemic options to improve care for this NSCLC population. I believe that implementing some of the above suggestions would make your important work even stronger.
Reviewer 2 Report
Comments and Suggestions for Authors
Methods - Section 2.3 - lines 152-3. Did both pathologists analyze all the specimens chosen for this study? If so, what was the concordance for the diagnoses? If not, could this division of labor have confounded the diagnoses?
Results - Section 3.1 - lines 176 - 9. Could have the more advanced stage of disease for patients allocated initially to MTT have confounded your interpretation of findings? Is it possible that earlier MTT availability to patients with less severe disease would have increased the cure rate?
Figures 2 and 3 - Could have the small sample size confounded your lack of finding a survival benefit, especially between 12 and 48 months ?
Discussion - Since citation #26 is of a similar design to your study, but their sample size was smaller, is it possible to combine the findings of the two studies to determine whether statistical significance might be found with a larger overall sample size for treatment intervention comparisons?
Reference 26 included analysis by sex. Since your sample size was larger than theirs, and you had a reasonable balance by sex in your study population, why didn't you include sex as a potential contributor to your research findings?
Round 2
Reviewer 1 Report
Comments and Suggestions for Authors
Thank you for the attention and consideration you have shown to my suggested revisions for your manuscript. It is evident that a significant amount of effort and thought has been directed towards the refining of your work, integrating the feedback provided during the peer review process. The resulting modifications demonstrate a thorough approach and significantly improve the rigor and overall quality of your manuscript. I look forward to witnessing the impact your research will undoubtedly have on the academic community.